# Immunometabolic Markers in a Small Patient Cohort Undergoing Immunotherapy

**DOI:** 10.3390/biom12050716

**Published:** 2022-05-18

**Authors:** Joshua Hofbauer, Andreas Hauck, Carina Matos, Nathalie Babl, Sonja-Maria Decking, Michael Rechenmacher, Christian Schulz, Sabine Regotta, Marion Mickler, Sebastian Haferkamp, Peter J. Siska, Wolfgang Herr, Kathrin Renner, Marina Kreutz, Annette Schnell

**Affiliations:** 1Department of Internal Medicine III, University Hospital Regensburg, 93053 Regensburg, Germany; joshua.hofbauer@stud.uni-regensburg.de (J.H.); carina.matos@ukr.de (C.M.); Nathalie.Babl@ukr.de (N.B.); sonja-maria.decking@ukr.de (S.-M.D.); michael.rechenmacher@ukr.de (M.R.); peter.siska@ukr.de (P.J.S.); wolfgang.herr@ukr.de (W.H.); Kathrin.renner-sattler@ukr.de (K.R.); marina.kreutz@ukr.de (M.K.); 2Department of Internal Medicine IV, Saarland University Hospital, 66421 Homburg, Germany; andreashauck1975@googlemail.com; 3Leibniz Institute for Immunotherapy (LIT), University Hospital Regensburg, 93053 Regensburg, Germany; 4Department of Internal Medicine II, University Hospital Regensburg, 93053 Regensburg, Germany; christian.schulz@ukr.de (C.S.); sabine.regotta@ukr.de (S.R.); 5Department of Dermatology, University Hospital Regensburg, 93053 Regensburg, Germany; marion.mickler@ukr.de (M.M.); sebastian.haferkamp@ukr.de (S.H.)

**Keywords:** PD-1, immunotherapy, PD-1+ monocytes, metabolism, hormones, immune cells, cholesterol, testosterone, free androgen index, hemoglobin

## Abstract

Although the discovery of immune checkpoints was hailed as a major breakthrough in cancer therapy, generating a sufficient response to immunotherapy is still limited. Thus, the objective of this exploratory, hypothesis-generating study was to identify potentially novel peripheral biomarkers and discuss the possible predictive relevance of combining scarcely investigated metabolic and hormonal markers with immune subsets. Sixteen markers that differed significantly between responders and non-responders were identified. In a further step, the correlation with progression-free survival (PFS) and false discovery correction (Benjamini and Hochberg) revealed potential predictive roles for the immune subset absolute lymphocyte count (rs = 0.51; *p* = 0.0224 *), absolute basophil count (rs = 0.43; *p* = 0.04 *), PD-1+ monocytes (rs = −0.49; *p* = 0.04 *), hemoglobin (rs = 0.44; *p* = 0.04 *), metabolic markers LDL (rs = 0.53; *p* = 0.0224 *), free androgen index (rs = 0.57; *p* = 0.0224 *) and CRP (rs = −0.46; *p* = 0.0352 *). The absolute lymphocyte count, LDL and free androgen index were the most significant individual markers, and combining the immune subsets with the metabolic markers into a biomarker ratio enhanced correlation with PFS (rs = −0.74; *p* ≤ 0.0001 ****). In summary, in addition to well-established markers, we identified PD-1+ monocytes and the free androgen index as potentially novel peripheral markers in the context of immunotherapy. Furthermore, the combination of immune subsets with metabolic and hormonal markers may have the potential to enhance the power of future predictive scores and should, therefore, be investigated further in larger trials.

## 1. Introduction

The discovery of immune checkpoints and the development of their specific inhibitors were celebrated as a major breakthrough in cancer therapy. In particular, blocking the inhibitory receptor PD-1 on immune cells and its ligand PD-L1 on both immune and tumor cells has been shown to be associated with an enhanced overall survival in metastatic disease of various tumor entities. However, only a limited patient cohort demonstrated a sufficient response to therapy [1]. Hence, there is a need to identify new checkpoints and predictive biomarkers in immunotherapy, with the objective of overcoming immune escape and resistance to treatment.

At present, researchers have elucidated primary resistance mechanisms including insufficient antigen immunogenicity, dysfunction of antigen presentation, irreversible T cell exhaustion, resistance to IFN-γ signaling, oncogenic signaling and the immunosuppressive tumor microenvironment that impedes upon antitumor immunity. Furthermore, an acquired resistance subsequent to the initial response has previously been discussed due to cancer immunoediting, the activation of other inhibitory pathways and re-exhaustion of reinvigorated T cells [2].

Biomarkers may guide how to navigate resistance; furthermore, the identification of new biomarkers may unmask still unknown therapeutic obstacles. To date, a range of individual biomarkers has been identified for different tumor entities and stages.

The high levels of microsatellite instability, deficiency in mismatch repair, high density of tumor-infiltrating lymphocytes, high PD-L1 expression on tumor and immune cells, absence of Galectin-3 and distinct composition of the gut microbiota were associated with response to treatment [2,3].

Other researchers found a predictive significance for routine blood tests before the onset of immunotherapy. Serum LDH, CRP levels, hemoglobin levels, leukocyte-to-lymphocyte ratio, neutrophil-to-lymphocyte ratio, monocyte count, basophils, eosinophils and absolute lymphocyte counts exhibited predictive power [4,5,6]. Furthermore, tumor- and macrophage-related factors such as CXCL5 or sCD163 and different monocyte subsets were shown to correlate with response [4,7,8,9].

None of the markers mentioned above securely identify responders; hence, the reasons as to why response rates are limited require further elucidation. In addition, there is an urgent need for the development of reliable biomarker combinations for different tumor entities. In particular, metabolic and hormonal markers in routine blood tests have scarcely been investigated in the context of immunotherapy and have only recently been considered [10,11,12,13,14]. Initial attempts to combine peripheral biomarkers have focused mainly on immune subsets or inflammatory markers [15,16,17,18,19,20,21].

We performed an exploratory study in a small patient cohort undergoing immunotherapy. Differential blood counts, blood serum markers, leukocyte subsets, checkpoint markers and metabolic markers were analyzed. In a first step, 16 peripheral target analytes associated with response to therapy after 6 months were identified. In a second step, correlating the 16 markers of interest with PFS suggested a potential predictive role for the free androgen index and PD-1+ monocytes, in addition to confirming well-established markers. Multiple correlation analysis revealed a strong inverse correlation between PD-1+ monocytes and hemoglobin. The combination of immune subsets with metabolic markers enhanced correlation with PFS.

The aim of this exploratory and hypothesis-generating study was to identify promising new target analytes in the context of immunotherapy, to emphasize the potential relevance of peripheral metabolic and hormonal markers and to discuss a possible combination of immune subsets with metabolic and hormonal markers in future predictive scores.

## 2. Materials and Methods

### 2.1. Patients

Our exploratory study focused on cancer patients receiving checkpoint immunotherapy with an anti-PD-1 antibody (Nivolumab (OPDIVO; Bristol-Myers Squibb, New York City, NY, USA) or Pembrolizumab (KEYTRUDA; MSD Merck Sharp & Dohme, Kenilworth, NJ, USA)) at the University Hospital Regensburg, Germany, from 2016 to 2018. Thirty-two patients fulfilled the criteria for analysis and were included. Exclusion criteria were treatments with other immunotherapies (anti-programmed-death-ligand (PD-L)1, anticytotoxic-T-lymphocyte-associated protein (CTLA)4, double checkpoint blockade), simultaneous radiotherapy, simultaneous treatment with chemotherapy, simultaneous treatment with a prednisolone equivalent to over 20 mg/day, follow-up not possible, sudden early death (response not evaluable) and no staging within 4 weeks before onset of treatment. All patients provided informed consent after being informed of the details of the study. The study (Z-2015-0589-7) was registered, approved by the Institutional Ethics Committee of the University of Regensburg (vote number 15-101-0267, October 2015) and was in accordance with the Declaration of Helsinki.

Response to therapy was defined as stable disease or disease regression lasting for at least 6 months. Follow-up occurred until November 2020 (57 months from start date). Blood was drawn as part of the clinical routine at the onset of immunotherapy. Peripheral blood samples were analyzed in the routine clinical hospital laboratory and by flow cytometry.

In a first step, 16 peripheral target analytes that differed significantly between responders and non-responders were identified. Normality tests were performed for each marker. For normally distributed data, we applied an unpaired t test (two-tailed). For normal distribution with significantly different variances, we applied an unpaired t test (two-tailed) with Welch’s correction. For cases exhibiting non-normal distribution, we conducted a Mann–Whitney test. Multiple testing was not corrected in this first step. In a second step, the 16 markers of interest were correlated with PFS (Spearman correlation). A false discovery rate correction was performed as proposed by Benjamini and Hochberg [22]. For the 7 markers correlating significantly with PFS, we created a biomarker map as described in the results section. Cut-off values for the different analytes between responders and non-responders were estimated by ROC curves. The datapoint on the ROC curve nearest to the upper left corner of the graph (highest sensitivity and specificity) was chosen. Furthermore, in some of the graphs, if 2 or more points were positioned near the upper left corner, the datapoint with the highest likelihood ratio was chosen. Multiple correlation analysis was corrected by Bonferroni testing. Significance was indicated as * *p* < 0.05, ** *p* < 0.01, *** *p* < 0.001 and **** *p* < 0.0001. All calculations were performed using GraphPad Prism 8 Software, https://statistikguru.de/rechner/adjustierung-des-alphaniveaus.html (Hemmerich, W. (2016)) (accessed on 13 May 2022) and https://www.sdmproject.com/utilities/?show=FDR (accessed on 13 May 2022).

Patient characteristics are summarized in Table 1. Seventy-five percent of the participants were males. The mean age was 67.78 (ranging from 34 to 85); the mean body mass index (BMI) was 24.19 (ranging from 17.6 to 35.35). There was no significant difference between the responders and non-responders for age and BMI. A total of 21.88% were on statin medication, and 21.88% were on prednisolone medication. Among the participants, 56.25% were treated with non-steroidal anti-inflammatory drugs (NSAIDs) at the onset of immunotherapy. A total of 37.5% of the primary tumors were non-small cell lung cancer (NSCLC); 25% were melanoma; 18.75% were head and neck squamous cell carcinoma (HNSCC), and 18.75% were other types. Twenty-five percent received immunotherapy as first-line therapy (87.5% of the first-line therapy occurred in the responder group). Concerning treatment, 31.25% were treated with Pembrolizumab and 68.75% with Nivolumab. Half developed adverse events, and 81.25% of these adverse events were documented in the responder group.

### 2.2. Clinical Blood Parameters, Blood Metabolites and Hormones

Clinical blood parameters were routinely drawn by the attending physician before the onset of immunotherapy. Serum was acquired by centrifugation for analysis of blood metabolites and hormones. All samples were analyzed in the central hospital laboratory.

### 2.3. Flow Cytometry

Blood samples were treated with a lysis buffer. Cells were stained with fluorochrome-conjugated antibodies following the manufacturer’s instructions. Cell populations were acquired by using a Fortessa flow cytometer (Beckton Dickinson, Franklin Lakes, NJ, USA), and the data were analyzed by using the FlowJo software (Tree Star, Ashland, OR, USA). The common gating strategy is shown in Appendix A (FACS Gating).

## 3. Results

Peripheral blood samples were acquired from each patient at the onset of immunotherapy. Differential blood counts, blood serum markers, leukocyte subsets, checkpoint markers and metabolic markers were analyzed. Six months after initiation of treatment, the response to therapy was evaluated, and the expression of the target analytes was compared between responders and non-responders. In this first step, we intended to perform an initial assessment in order to identify relevant markers for further analysis. Multiple testing was not corrected.

### 3.1. Identification of Target Analytes in Responders and Non-Responders

Consistent with preceding trials, higher baseline lymphocyte counts [23,24], basophil counts [25], low-density lipoprotein (LDL), high-density lipoprotein (HDL) [10,26,27,28] and hemoglobin (HB) levels [6] were found within the responder patients, while enhanced levels of CRP [5,29,30,31,32] and CD33high CD11b+ monocytes, which may resemble myeloid- derived suppressor cells (MDSC) [33,34], were related to treatment failure (Table 2 and Appendix A).

No novel findings were reported after analyzing the remaining differential blood count parameters, lymphocyte subsets (T cells, B cells, plasmacytoid dendritic cells (pDC) and natural killer (NK) cells) and the expression of checkpoint markers on lymphocytes (FACS gating in Appendix A and the tables in Appendix A).

After investigating the myeloid subsets, we identified three populations of interest: PD-1+ monocytes, PD-1+ granulocytes and HLA-DR+ CD16+ (medium) monocytes. All three populations were associated with treatment failure in patients receiving immunotherapy (Table 3, Figure 1).

PD-1-positive myeloid cells could play an important role in immunotherapy, as the targeted deletion of PD-1 in mice myeloid cells induced antitumor immunity [35]. Furthermore, an enhanced expression of PD-1 on monocytes in renal cell carcinoma patients was associated with a lower cancer-specific survival [36]. A decrease in CD16+ HLA-DR+ monocytes, accompanied by a significant increase in overall survival, was observed in recurrent glioblastoma after post-surgical Pembrolizumab, but only in patients who had also received neoadjuvant anti-PD-1 immunotherapy before surgery [9]. All the other investigated checkpoint markers on monocytes did not differ between responders and non-responders (Appendix A). In the process of analyzing the myeloid markers CD33 and CD11b on leukocytes, we incidentally found that a CD33- CD11b- subset was associated with response. The subset resembled a lymphocyte population in the FSC/SSC plot when back-gated (Figure 2, Table 4).

Furthermore, we paid special attention to the cellular expression of the metabolic markers CD39 and CD147. CD39 belongs to the family of Ectonucleotidases and has been linked to immunosuppression and tumor metastasis. Its impact on immunotherapy is controversial and has previously been discussed in the literature. [37,38,39]. CD147 has been associated with immunosuppression and described as a negative regulator of antitumor responses mediated by CD8+ tumor-infiltrating lymphocytes [40,41,42,43]. The expression of both markers on CD19- lymphocytes was associated with treatment failure; interestingly, the expression of CD39 on B cells was associated with response (Figure 2, Table 4). We did not find any relevant differences for other serum metabolites (Appendix A).

As gender-specific differences have been discussed in immunotherapy research [14], we also analyzed baseline hormonal metabolites at treatment onset and found an association with response for testosterone and the free androgen index in male patients (Figure 3, Table 5). An identical trend was found in female patients at lower levels; however, due to the low proportion of female patients, a valid statement could not be made. Immunotherapy has been described as more effective in males by trend in comparison to females [14], possibly due to an increased antigenicity in male cancers and a less stimulated basic immune response. This may also be the case for a unisex male population with different testosterone levels. We did not find any relevant differences for other hormonal metabolites (Appendix A).

In summary, we revealed 16 peripheral markers associated with the response to immunotherapy. Although multiple testing was not corrected in this first step, it is noteworthy that the differences between responders and non-responders were highly significant for the absolute lymphocyte count and LDL (***), followed by the absolute basophil count and the free androgen index (**).

### 3.2. Correlation with PFS-Introduced PD-1+ Monocytes and the Free Androgen Index as Potentially Novel Markers in Immunotherapy

Having defined 16 immunometabolic subsets and proteins of interest, we correlated them with progression-free survival (PFS). Follow-up occurred over 57 months from the start date. A false discovery rate correction was performed as proposed by Benjamini and Hochberg (Table 6, Appendix A) [22].

In the group of familiar markers in immunotherapy, as expected, the absolute lymphocyte and basophil counts, LDL levels and HB levels correlated positively with PFS. CRP correlated negatively with PFS. Furthermore, we revealed a positive correlation for the free androgen index with PFS in male patients and a negative correlation between PFS and the PD-1+ monocyte subset.

The most powerful significance values were assigned to the metabolic markers LDL and the free androgen index in addition to the lymphocyte subset, followed by CRP. Testosterone is predominantly bound to blood proteins and, therefore, may not correlate significantly with PFS in contrast to the biologically active free testosterone. Noticeably, in our patient population, HDL did not reveal a significant coherence with PFS, although LDL exhibited one of the most powerful significances. LDL has been reported to enhance lymphocytes in healthy humans [44,45] and might have a completely different impact on the immune system than HDL. All the other monocyte subsets, other than the PD-1+ subset and the CD39- and CD147-expressing lymphocytes, did not show a significant correlation with PFS.

### 3.3. Combining Immune Subset Markers with Metabolic Markers Enhanced Correlation with PFS

As the correlations between the single analytes and PFS were only moderate, we assumed that a combination of the markers could increase significance. Using ROC curves, we estimated cut-off values for each analyte (Appendix A) and assigned value ranges to treatment response or failure. The values for each patient were entered into a biomarker heatmap, and a ratio was formed (Figure 4, data in Appendix A). The green fields represent a value within the responder range, and the red fields represent a value within the non-responder range for each analyte. Black fields represent missing data. The ratio was formed by dividing the red fields by the total fields (green and red fields), generating values between 0 and 1. Values larger than 0.5 indicated a higher number of analytes in the non-responder range (red fields), and values smaller than 0.5 indicated a higher number of analytes in the responder range (green fields).

The correlation of the immune subset biomarker map ratio with PFS showed a moderate inverse correlation to (Spearman rs = −0.56, *p* = 0.0010 *** and 95% confidence interval −0.76 to −0.25). Combining the immune subsets with the metabolic markers strongly enhanced inverse correlation to (Spearman rs = −0.74, *p* ≤ 0.0001 **** and 95% confidence interval −0.87 to −0.51). Statistics are reported in Appendix A.

Therefore, we suggest that a combination of immune subsets and metabolic markers should be evaluated for future predictive scores.

### 3.4. Multiple Correlation Analysis Revealed a Strong Inverse Correlation between PD-1+ Monocytes and Hemoglobin

In a third step, we performed a multiple correlation analysis to search for coherences between the markers of interest (Figure 5, Appendix A). Due to the high number of tests, Bonferroni correction was conducted. In addition to a strong positive correlation between LDL and the free androgen index (rs = 0.73, *p* = 0.0000745 **** and corrected *p* = 0.00298 **) and between lymphocytes and basophils (rs = 0.61, *p* = 0.0003 *** and corrected *p* = 0.012 *), an intriguing inverse correlation was revealed between PD-1+ monocytes and hemoglobin (rs = −0.755, *p* = 0.0000769 **** and corrected *p* = 0.003076 **).

## 4. Discussion

Thus far, relying on single biomarkers has not been suitable for achieving reliable predictions of response rates in tumor patients undergoing immunotherapy. Furthermore, tumor biology often changes during disease progression. Tumor biopsies are necessary for the re-evaluation of therapeutical options, but they regularly require invasive interventions. The acquisition of predictive markers in peripheral blood samples would be a significant relief for many patients.

Initial attempts to combine peripherally available markers in addition to clinical parameters have recently been initiated, inter alia by integrating the Glasgow prognostic score, the neutrophil-to-lymphocyte ratio, the systemic immune–inflammation index, LDH, the lung immuno-oncology prognostic score (LIPS)-3, plasma metabolites (hippuric acid, butyryl carnitine, cystine and glutathione) and cell surface markers in well-defined patient populations [15,16,17,18,19,20,21].

Considering the summary of our findings in this exploratory approach, we suggest that the integration of metabolic and hormonal markers into immune-subset-dominated scores should be further evaluated and might have the potential to enhance their predictive power. In addition, a detailed investigation of the very heterogenous myeloid subsets could potentially be promising.

In our work, in addition to well-established markers, two potentially novel peripheral markers emerged: the free androgen index and the PD-1+ monocyte subset.

The free androgen index (testosterone/sexual hormone binding globulin × 100) correlated with the PFS in male patients and was also strongly associated with LDL, possibly due to the fact that testosterone is synthesized from cholesterol under the control of the gonadotropin LH [46]. However, cholesterol delivery is not only provided by LDL, i.e., Leydig cells can produce cholesterol by de novo synthesis. It has been demonstrated that patients on treatment with a PSCK-9 inhibitor with very low LDL levels did not develop adverse effects concerning steroid or gonadal hormones [47,48]. Alternatively, the effect may be based upon cancer cachexia, which is also associated with an impaired response to immunotherapy, on the one hand, due to a lack of substrate and, on the other hand, presumably due to the effect of other inflammatory metabolites and cytokines [49,50,51]. Furthermore, vice versa, an initial testosterone deficiency or testosterone supplementation may secondarily alter the patient’s lipid profile [52,53].

Contrarily, in cancer patients suffering from the metabolic syndrome, we would rather expect high LDL levels combined with low testosterone levels, so these markers are not inevitably positively correlated with each other in every patient population. Furthermore, in context with the metabolic syndrome, HDL might have a noticeably stronger impact than LDL [54,55].

We believe that metabolism regarding lipids and hormones is a very complex and important topic for the efficacy and understanding of immunotherapy and may also explain some of the paradoxical findings for cholesterol and immunotherapy in the literature.

Further research must elucidate if (a) both markers—LDL and the free androgen index—have a biological impact on immunotherapy, (b) if only one of them has an impact or (c) a third unknown factor is at play. LDL, the free androgen index and the absolute lymphocyte count revealed the strongest significance for correlation with PFS in this patient cohort. Correlating with the absolute lymphocytes as well (although not significantly after Bonferroni correction), LDL may potentially be a very powerful single marker for predicting response to immunotherapy in well-defined populations. Several studies underline that hypercholesterolemia may be associated with response to immunotherapy [10,26,27,28]; however, the individual roles of LDL and HDL are still not clear. Contrariwise, in a population of melanoma patients, oxidized lipoproteins have also been linked to therapeutic failure [56].

Recently, two papers have been published showing that testosterone has been identified as a novel crucial factor for immunotherapy in prostate and bladder carcinoma by directly diminishing CD8+ T cell function via the androgen receptor [57,58].

Similarly, like cholesterol, which has been demonstrated to induce exhaustion in CD8+ cells in the tumor environment [59], high serum levels in the peripheral blood seem to be associated with response, and testosterone suppresses CD8+ cells via the androgen receptor in the tumor environment, but high serum levels may nevertheless be favorable. Lower testosterone serum levels might enhance the androgen receptor sensitivity on T cells and also the conversion rate of testosterone to dihydrotestosterone, which enhances binding to the androgen receptor, and the individual tumor tissue may play an important role. Furthermore, downregulation of testosterone may, on the other hand, diminish the function of other immune cells. In mice, a reduction in testosterone impaired antitumor neutrophil function [60].

For some tumor entities, however, immunotherapy has been described as more effective in males in comparison to females by trend [14], possibly due to an increased antigenicity in male cancers and a less stimulated basic immune response. This may also be the case for a unisex male population with different testosterone levels. Recently, sex-specific hormone changes during immunotherapy were demonstrated in a small population of patients with metastatic renal cell carcinoma. A significant negative association between the LH/FSH ratio and progression-free survival was revealed [61]. It was also shown that glucose metabolism, obesity-associated inflammation and cancer immunity differ between male and female tumor patients [62]. In principle, it is conceivable that hormone-controlled gender differences could also be evident in the same sex, in keeping with the patient’s individual hormone levels. 

The androgen pathway and CD147 might also have a mutual impact on each other. CD147 has been reported to modulate androgen receptor activity in prostate cancer cells. Being a chaperone to some monocarboxylate-transporter isoforms, CD147 is also involved in glucose metabolism. CD147 overexpression contributed to the metabolic transformation of tumors by accelerating aerobic glycolysis and lactate efflux [63,64,65]. Furthermore, CD147 is a potent inducer of extracellular matrix metalloproteinases [66], and the expression of CD147 on regulatory T cells was associated with an enhanced regulatory function [40,42,67]. The CD147+ CD19- lymphocyte population in our data perhaps represents regulatory T cells. Furthermore, CD147 has also been described as a negative regulator of antitumor responses mediated by CD8+ tumor-infiltrating lymphocytes [43].

In summary, the free androgen index may have the potential to serve as a predictive marker in well-defined patient populations. However, there are many clinical obstacles to overcome, i.e., with regard to gender, age, tumor entity and the patient’s metabolic state. The relevance of testosterone substitution could be evaluated in non-androgen-dependent tumors, especially as this also might alter the patient’s lipid profile. 

The PD-1+ monocyte subset was especially intriguing, as it revealed a strong inverse correlation with hemoglobin, in addition to correlating negatively with PFS. Enhanced expression levels of PD-1 on monocytes were associated with a lower cancer-specific survival in renal cell carcinoma patients [36]. The targeted deletion of PD-1 in mice myeloid cells induced antitumor immunity [35]. Furthermore, in a cohort of metastatic urothelial carcinoma patients, treatment with a PD-1 inhibitor decreased the frequency of PD-1+ monocytic MDSCs by trend [9].

The strong inverse correlation with hemoglobin may be due to the monocytes scavenging erythrocytes and their components [68]. Iron supplementation has been reported to interfere with immunotherapy in mice [69]. Perhaps monocyte overload with debris and iron could induce PD-1 upregulation. Alternatively, cytokines such as IL-6, IL-1 and TNF-alpha and metabolites associated with anemia of chronic disease could modulate monocyte function and lead to an upregulation of PD-1. The oral hypoxia-inducible factor prolyl hydroxylase inhibitor Roxadustat corrected inflammation-induced anemia in chronic kidney disease [70]. Perhaps medication with Roxadustat could optimize the efficacy of immunotherapy in some patients. The value of substituting erythrocyte concentrates and treating anemia in immunotherapy has to be clarified.

The levels of HLA-DR+ CD16+ monocytes varied between responders and non-responders; however, the correlation with PFS was not significant. CD16+ monocytes have been linked to inflammation, angiogenesis and the production of reactive oxygen species [71]. As already mentioned above, a similar population has been reported in glioblastoma patients. A decrease in CD11c+ CD14+ CD16+ HLA-DR high monocytes, accompanied by a significant increase in overall OS, was observed in recurrent glioblastoma after post-surgical Pembrolizumab, but only in patients that had also received neoadjuvant anti-PD-1 immunotherapy before surgery [9]. Furthermore, it was postulated that CD16+ monocytes produce high amounts of IL-10 [72]. CD16+ monocytes were also observed in ovarian cancer progression and have been linked to immunosuppression [73].

We also did not see any differences for HLA-DR+ CD16- monocytes between responders and non-responders. CD14+ HLA-DR+ CD16- monocytes were associated with the response to checkpoint therapy in melanoma patients [7]. Furthermore, a high number of PD-L1+ monocytes was related to a shorter OS, and HLA-DR low monocytes were linked to a poor response in immunotherapy [8,74]. Here, we also did not observe relevant differences between the treatment groups.

Moreover, we observed a strong correlation between lymphocytes and basophils, possibly due to an enhanced release through the bone marrow, especially as the composition of the lymphocyte subsets did not differ between responders and non-responders. LDL has been reported to enhance lymphocytes in healthy humans [44,45], and testosterone directly induced IL-33 expression through mast cells, driving the generation of both innate lymphoid cells and basophils [75].

## 5. Conclusions

In summary, in addition to validating well-established markers in our exploratory and hypothesis-generating study, we also identified PD-1+ monocytes and the free androgen index as potentially novel peripheral markers in the context of immunotherapy. Further research should be conducted to clarify whether the free androgen index may have the potential to serve as a predictive marker in well-defined patient populations. There are still many clinical obstacles to overcome, i.e., with regard to gender, age, tumor entity and the patient’s metabolic state. The relevance of testosterone substitution could be evaluated in non-androgen-dependent tumors, especially as this also might beneficially alter the patient’s lipid profile.

Furthermore, the strong inverse correlation between PD-1+ monocytes and hemoglobin may also imply potential treatment options. Pathophysiological interdependencies, substitution and medical treatment of anemia should be investigated in the context of immunotherapy. Moreover, the combination of immune subsets with metabolic and hormonal markers may be less prone to failure and, therefore, have the potential to enhance the power of future predictive scores. Combined scores should be validated for well-defined patient populations and further investigated in larger trials. 

## Figures and Tables

**Figure 1 biomolecules-12-00716-f001:**
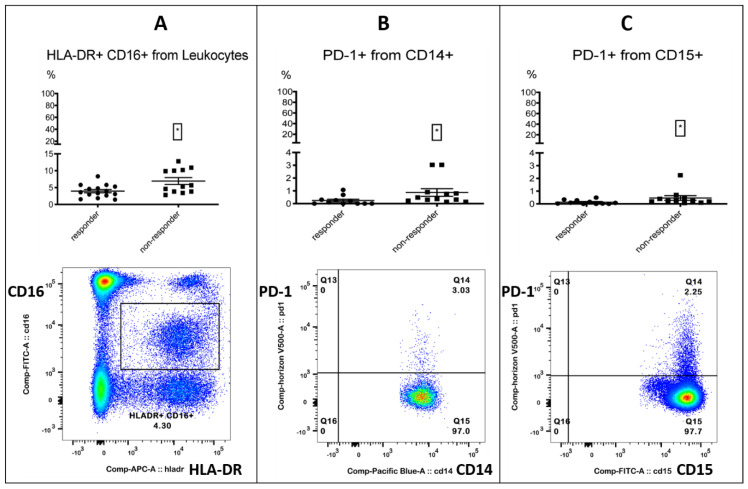
Myeloid cells. Three myeloid subsets were associated with treatment failure. The top row shows the % positive from the parent population for responders and non-responders. Left column: responders; right column: non-responders. The bottom row shows the FACS gating. **Left** (**A**)**:** HLA-DR+ CD16+ (medium) monocytes % from leukocytes. **Middle** (**B**)**:** PD-1+ monocytes % from CD14+ monocytes. **Right** (**C**)**:** PD-1+ granulocytes % from CD15+ granulocytes. Significance was indicated as * *p* < 0.05.

**Figure 2 biomolecules-12-00716-f002:**
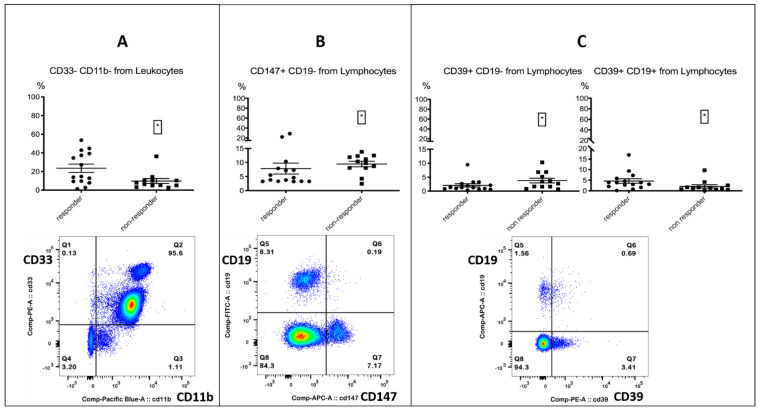
Lymphocytes. The top row shows the % positive from the parent population for responders and non-responders. Left column: responders; right column: non-responders. The bottom row shows the FACS gating. **Left** (**A**)**:** CD33- CD11b- population % from leukocytes, associated with response. **Middle** (**B**)**:** CD147+ CD19- lymphocytes % from lymphocytes, associated with treatment failure. **Right** (**C**)**:** CD39+ CD19- lymphocytes % from lymphocytes, associated with treatment failure and CD39+ B cells, associated with response. Significance was indicated as * *p* < 0.05.

**Figure 3 biomolecules-12-00716-f003:**
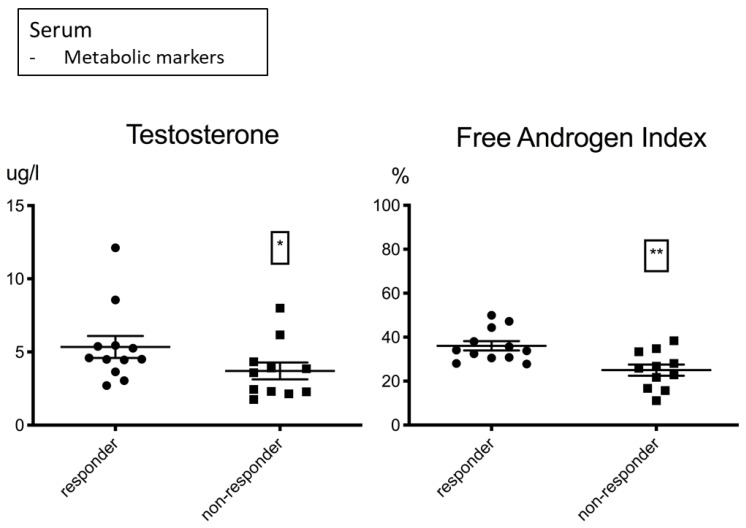
Testosterone µg/L (**left**) and the free androgen index % (**right**) shown for male responders and non-responders. Higher levels were associated with response to treatment. Significance was indicated as * *p* < 0.05, ** *p* < 0.01.

**Figure 4 biomolecules-12-00716-f004:**
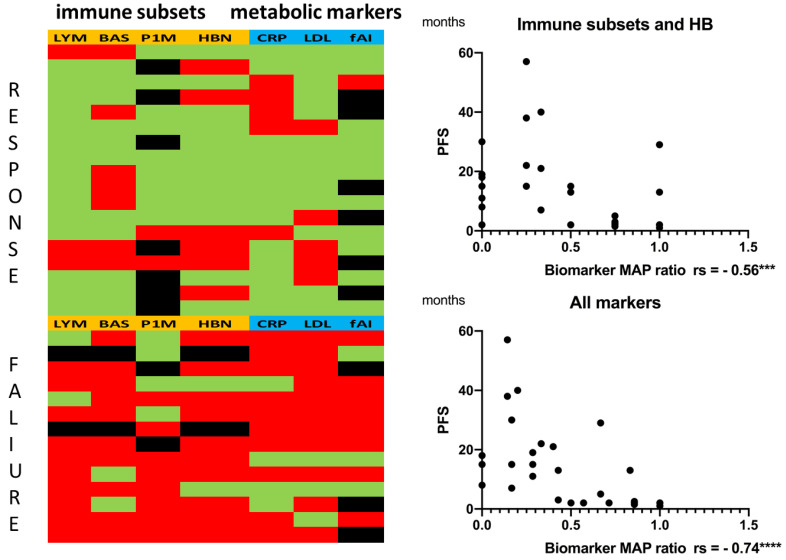
(**Left biomarker heatmap**) Immune subsets and hemoglobin (orange): LYM = absolute lymphocyte counts, BAS = absolute basophil counts, P1M = PD-1+ monocytes, HBN = hemoglobin. Metabolic markers (blue): CRP = C-reactive protein, LDL = low-density lipoprotein, fAI = free androgen index. **Green:** value within range, responder. **Red:** value within range, non-responder. **Black:** no data (fAI only for male patients: 8 patients were female). A ratio (biomarker map ratio) was formed by dividing the red fields by the total fields (green and red fields). (**Right**) Correlation between the biomarker map ratio and PFS: (**Top**) immune subsets only (with hemoglobin). (**Bottom**) immune subsets combined with metabolic markers. Statistics are reported in Appendix A. Significance was indicated as *** *p* < 0.001 and **** *p* < 0.0001.

**Figure 5 biomolecules-12-00716-f005:**
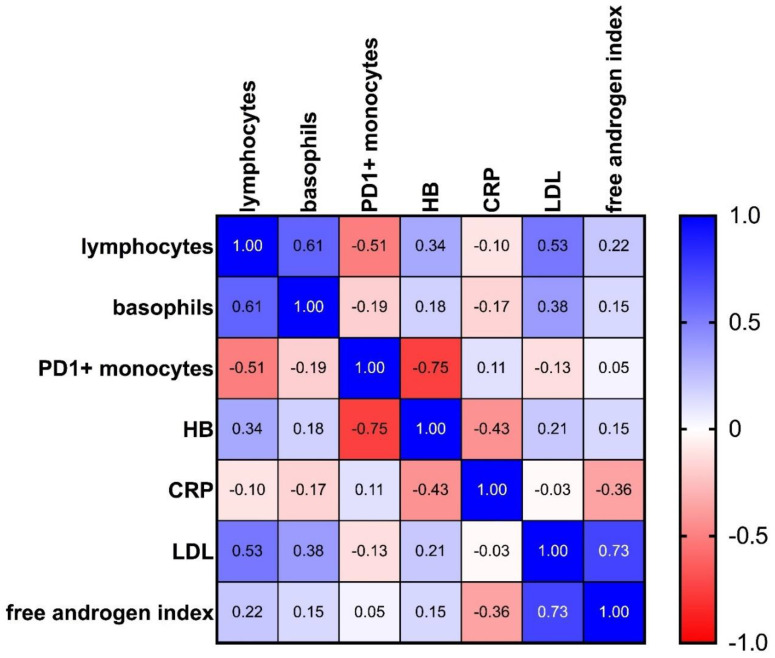
Heatmap showing multiple correlation analysis for the target analytes: 1.0 (blue) stands for a perfect positive correlation; 0 (white) means no correlation, and −1.0 (red) stands for a perfect negative correlation. Values between 0 and 1 indicate analytes increasing or decreasing together; values between 0 and −1 indicate that one analyte increases as the other decreases (see also Appendix A).

**Table 1 biomolecules-12-00716-t001:** Overview of patient characteristics.

Variables	Patients (32 = 100%)	Responders (18 = 56.25%)	Non-Responders (14 = 43.75%)
Female	8 (25%)	6 (18.75%)	2 (6.25%)
Male	24 (75%)	12 (37.5%)	12 (37.5%)
Age (years)	64.78 (34–85)	67.94 (46–85)	60.71 (34–82)
Body mass index (kg/m^2^)	24.19 (17.6–35.35)	25.35 (18.8–35.35)	22.86 (17.6–28.10)
Statin	7 (21.88%)	3 (9.38%)	4 (12.5%)
Prednisolone ≤ 20 mg	7 (21.88%)	5 (15.63%)	2 (6.25%)
NSAID	18 (56.25%)	8 (25%)	10 (31.25%)
**Primary tumor**			
NSCLC	12 (37.5%)	5 (15.63%)	7 (21.88%)
Melanoma	8 (25%)	8 (25%)	0 (0%)
HNSCC	6 (18.75%)	2 (6.25%)	4 (12.5%)
Others	6 (18.75%)	3 (9.38%)	3 (9.38%)
Previous treatments			
<1	8 (25%)	7 (21.88%)	1 (3.13%)
≥1	24 (75%)	11 (34.38%)	13 (40.63%)
Pembrolizumab	10 (31.25%)	9 (28.13%)	1 (3.13%)
Nivolumab	22 (68.75%)	9 (28.13%)	13 (40.63%)
Adverse events	16 (50%)	13 (40.63%)	3 (9.38%)

The number of patients and their results in percentages are shown for the entire study population, responders and non-responders. For age (years) and body mass index (kg/m^2^), the mean is shown with minimum and maximum. NSAID = non-steroidal anti-inflammatory drugs; NSCLC = non-small cell lung cancer; HNSCC = head and neck squamous cell carcinoma.

**Table 2 biomolecules-12-00716-t002:** Already familiar target analytes in immunotherapy.

Immune Subset	Standard Value or % Population	Responders	Non-Responders	Significance (*p*)
Absolute lymphocyte counts	(1.18–3.74/nL)	1.49/nL (±0.66)	0.74/nL (±0.26)	0.0002 ***
Absolute basophil counts	(0.01–0.08/nL)	0.03/nL (0.02/0.05)	0.02/nL (0.01/0.02)	0.0071 **
LDL	(<100 mg/dL)	131.5 mg/dL (108.75/165.5)	95 mg/dL (75.75/116)	0.0009 ***
HDL	(40–60 mg/dL)	50 mg/dL (40/67.50)	39.5 mg/dL (31/51.50)	0.0253 *
HB	(11.2–15.7 g/dL)	12.94 g/dL (±1.48)	11.29 g/dL (±1.83)	0.0109 *
CRP	(<5 mg/L)	5.25 mg/L (2.9/13.58)	11.25 mg/L (7/35.5)	0.0162 *
CD33high CD11b+ monocytes	% of leukocytes	1.89% (1.61/4.02)	4.94% (2.74/7.0)	0.0238 *

Depending on the normality tests, the mean with standard deviation (±) or median with 25%/75% percentile is shown. Statistics are reported in Appendix A. Multiple testing was not corrected. Significance was indicated as * *p* < 0.05, ** *p* < 0.01, *** *p* < 0.001.

**Table 3 biomolecules-12-00716-t003:** Three myeloid subsets associated with treatment failure.

Myeloid Subsets	% from Population	Responders	Non-Responders	Significance (*p*)
PD-1+ granulocytes	% CD15+ granulocytes	0.08% (0/0.28)	0.24% (0.19/0.44)	0.0266 *
PD-1+ monocytes	% CD14+ monocytes	0.1% (0/0.31)	0.43% (0.25/0.88)	0.0119 *
HLA-DR+CD16+	% leukocytes	3.62% (2.68/5.31)	5.49% (3.84/10.15)	0.0291 *

Depending on the normality tests, the mean with standard deviation (+/−) or median with 25%/75% percentile is shown. Statistics are reported in Appendix A. Multiple testing was not corrected. Significance was indicated as * *p* < 0.05.

**Table 4 biomolecules-12-00716-t004:** Lymphocyte subsets and metabolic markers.

Lymphocyte Subsets	% from Population	Responders	Non-Responders	Significance (*p*)
CD147+ CD19-	% from lymphocytes	4.61% (3.32/7.9)	10.1% (8.04/12.2)	0.0376 *
CD39+ CD19-	% from lymphocytes	1.3% (0.85/2.69)	3.07% (1.62/6.39)	0.0481 *
CD39+ CD19+	% from lymphocytes	3.88% (2.06/5.85)	1.3% (0.71/2.36)	0.0246 *
CD33- CD11b-	% from leukocytes	14.7% (9.34/39.3)	7.25% (5.1/10.55)	0.0321 *

Depending on the normality tests, the mean with standard deviation (±) or median with 25%/75% percentile is shown. Statistics are reported in Appendix A. Multiple testing was not corrected. Significance was indicated as * *p* < 0.05.

**Table 5 biomolecules-12-00716-t005:** Hormonal metabolites.

Hormonal Metabolites	Responders	Non-Responders	Significance (*p*)
Testosterone	4.55 µg/L (3.84/5.43)	3.58 µg/L (2.28/4.33)	0.0317 *
Free androgen index	36.08% (+/−7.4)	25.02 % (+/−8.48)	0.0031 **

Depending on normality tests, the mean with standard deviation (+/−) or median with 25%/75% percentile is shown. Statistics are reported in Appendix A. Multiple testing was not corrected. Data shown for males. Significance was indicated as * *p* < 0.05, ** *p* < 0.01.

**Table 6 biomolecules-12-00716-t006:** Target analyte correlations with PFS.

Target Analyte	Spearman rs	Significance (*p*)	Corrected *p*
Lymphocytes	0.51	0.0039 **	0.0224 *
Basophils	0.43	0.0175 *	0.04 *
CD33 high+ CD11b+	−0.28	0.1648	0.1758
CD33- CD11b-	0.31	0.1168	0.1335
HLADR+ CD16+	−0.43	0.0244 *	0.0471 (*)
PD-1+ granulocytes	−0.40	0.0662	0.0883
PD-1+ monocytes	−0.49	0.0166 *	0.04 *
Hemoglobin	0.44	0.0148 *	0.04 *
CD147+ CD19-	−0.36	0.0652	0.0883
CD39+ CD19-	−0.19	0.3369	0.3369
CD39+ CD19+	0.33	0.0918	0.1130
CRP	−0.46	0.0088 **	0.0352 *
LDL	0.53	0.0021 **	0.0224 *
HDL	0.39	0.0265 *	0.0471 (*)
Testosterone	0.45	0.0317 *	0.0507
Free androgen index	0.57	0.0042 **	0.0224 *

Target analytes were correlated with PFS. A false discovery rate correction was performed as proposed by Benjamini and Hochberg (corrected *p*). **Blue:** positive correlation with PFS. **Red:** negative correlation with PFS. **Grey**: no significant correlation. Significance was indicated as * *p* < 0.05, ** *p* < 0.01.

## Data Availability

All relevant data are available as Appendix A.

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
