# Peer review of "Immunometabolic Markers in a Small Patient Cohort Undergoing Immunotherapy"

_biomolecules, 2022, doi:10.3390/biom12050716_

Round 1
Reviewer 1 Report
The authors identified PD1+ monocytes and free androgen index as the predictive markers for immune check point immunotheraoy. However, their results showed that LDL is the best marker for the immunotherapy (p = 0.0009). p-value of LDL is smaller than that of PD-l+ monocytes (0.0119). Moreover, the authors described that testosterone is synthesized from cholesterol. Therefore, I wnder why the authors did not describe about LDL as the most useful predictive marker in Abstract. LDL can be measured very easily. Moreover, if the authors intended to emphasize the free androgen index, I recommend to describe more precisely how androgen specifically impacts on the immune check point immunotherapy insted of general effect on immune cell (leucocytes). Since the Title syas immunometabolic markers, then underlying mechanisms of identified metabolic markers in the immune check point immunotherapy should be described more clerary for the readers.
Reviewer 2 Report
The exploratory and hypothesis-generating study's objective was to identify novel target analytes in the context of immunotherapy, to emphasize the potential relevance of peripheral metabolic and hormonal markers, and to discuss the possibility of combining immune subsets with metabolic and hormonal markers in future predictive scores. The manuscript is well structured and well discussed. However, some points should be checked and corrected before its acceptance in this journal.
Therefore, according to my comments, I recommended the publication of the paper after minor revision.
- The abstract is not clear. Please add the aim and objective of the MS.
- The study's background should be clearly stated. Describe the introduction and review of the work.
- Please provide in the conclusion section. The authors should add the significance of this research and its potential practical application.
- The MS English needs to be improved. The article's English must be carefully checked for grammatical errors.
Reviewer 3 Report
The authors presented their exploratory study with peripheral blood samples of cancer patients undergoing immunotherapy, aiming to identify potential new target analytes for therapy outcome prediction. This is a very significant topic. The authors also got very interesting findings, especially the combination of immune subsets and metabolic markers as predictive scores for immunotherapy. However, it would be better if the authors could clarify the following points:
- The authors identified 16 peripheral target analytes that were significantly different between responders and non-responders. However, we I used the raw data in Supplemental Data 4 to calculate the average and p value (two tailed t-test) of each analytes, my results were very different from the authors'. For example, when compare the CRP between responders and non-responders, the p value I got from the data was 0.2218, while the authors' p value was 0.0162. My result indicated there was no significant difference of CRP between responders and non-responders, which is different from the authors' conclusion. Could the author clarify how they calculate the averages and p values?
- The authors mentioned they used ROC to estimate the cut off value of each analytes. But no details were provided. Could the authors clarify how they determine the cut off based on ROC?
- In section 3.3, the authors got the conclusion that there was a high correlation between PFS and biomarker MAP ratio. But the biomarker heatmap had a lot of black NO DATA points. I am wondering if the NO DARA points would potentially alter the correlation.
- Could the authors better organize the supplemental data?
Round 2
Reviewer 1 Report
As this reviewer pointedout earlier and the authors described in the revised version, LDL is a best predictive marker for the ICI therapy. the LDL measurement is routine labo test, but not androgen. Even if the author's result is correct, there is little point in measuring androgens. They explained that the product of LDL is androgen and that it is related, but even so, LDL measurement is easier to measure and commonly used. If the authors demonstrate that impact of androgen level is superior to LDL and underlying mechanisms is presented, I will be convinced. Moreover, the androgen level is dependent on the patient's age and it may be difficult to use for female patients. Thus, androgen as a predictive marker for the ICI treatment is difficult for common use and shold be further evaluated. In conclusion, even in this revise version, there is an impression that the title of this manuscript is forcibly immunometabolic, and therefore this revised version did not reach the publication level. In addition, since It is a small cohort, data corresponding to the values of LDL and androgen/free androgen index are can be presented to confirm the author's conclusion. Furthermore, In Table 6, I wonder corrected significance of lymphocytes, LDL and free androgen index are same as 0. 0224, although their significance (P) are different. Are they correct?
Reviewer 3 Report
The manuscript has been improved considerably. The authors have modified the method section and make it easier for readers to understand the data analysis process. Therefore I recommenf publication.
Author Response
Thank you very much for your feedback and help!